# The effects of ischemia during rest intervals on strength endurance performance

**Robert Trybulski**[1,2☯], **Marta Bichowska**[3☯], **Rafal Piwowar**[4☯], **Anna Pisz**[5☯], **Michal Krzysztofik**[6☯], **Aleksandra Filip-Stachnik**[6☯], **Krzysztof Fostiak**[3☯], **Piotr Makar**[3☯], **Michal Wilk**[5,6☯]*

1 Provita Zory Medical Center, Zory, Poland, 2 Department of Medical Sciences, The Wojciech Korfanty School of Economics, Katowice, Poland, 3 The Academy of Physical Education and Sport in Gdansk, Gdansk, Poland, 4 Health Fit, Zory, Poland, 5 Faculty of Physical Education and Sport, Charles University, Prague, Czech Republic, 6 Institute of Sport Sciences, Jerzy Kukuczka Academy of Physical Education in Katowice, Katowice, Poland

☯ These authors contributed equally to this work.
* m.wilk@awf.katowice.pl

**Data Availability Statement:** All relevant data are within the manuscript and its Supporting information files.

## Abstract

### Background

The study aimed to evaluate the effects of ischemia used during the rest periods between successive sets on maximal number of performed repetitions, time under tension and bar velocity during the bench press exercise.

### Methods and materials

Thirteen healthy resistance trained men volunteered for the study (age = $28.5 \pm 7.1$ years; body mass = $87.2 \pm 8.6$ kg; bench press 1RM = $143.1 \pm 20.7$ kg; training experience = $11.0 \pm 6.9$ years). In experimental protocol the subjects performed 5 sets of bench press exercise at 70%1RM with maximal number of repetitions in each and with 5 minutes rest periods between each set. During the ischemia condition occlusion with 80% arterial occlusion pressure (AOP) was applied using a 10 cm wide cuff, before the first set of the bench press exercise and during all rest periods between sets (for 4.5 minute). During the control condition no ischemia was applied.

### Results

The two-way repeated measures ANOVA showed a statistically significant interaction effect for time under tension ($p = 0.022$; $\eta^2 = 0.20$). However, the results did not show a statistically significant interaction effect for peak bar velocity ($p = 0.28$; $\eta^2 = 0.10$) mean bar velocity ($p = 0.38$; $\eta^2 = 0.08$), and for number of performed repetitions ($p = 0.28$; $\eta^2 = 0.09$). The post hoc analysis for interaction showed significantly shorter time under tension for ischemia condition compared to control in set 1 ($p < 0.01$). The post hoc analysis for main effect of condition revealed that time under tension was significantly shorter for ischemia compared to control condition ($p = 0.04$).

**Funding:** The study was supported by the statutory research of the Jerzy Kukuczka Academy of Physical Education in Katowice, Poland. The study was also funded by the Grant Agency of Charles University through a grant awarded to MK and MW (PRIMUS/22/HUM/019). The funders had no role in study design, data collection and analysis, decision to publish, or preparation of the manuscript.

**Competing interests:** The authors have declared that no competing interests exist.

## Conclusion

The results of this study indicate that ischemia intra-conditioning does not increase strength-endurance performance as well as bar velocity during bench press exercise performed to muscle failure.

## Introduction

Ischemia is a method of restricting blood flow through the special cuffs which can be used in the limbs and can be applied during the rest as well during the different types of physical activity [1]. So far, much attention has been devoted to the use of ischemia during resistance exercise. Ischemia induced before or during effort increase exercise physical performance as well as stimulates physiological responses such as improvement of metabolic efficiency (attenuating ATP depletion), also glycogen depletion and the lactate production [2–11]. Further ischemia causes changes in systemic VO2, in the deoxygenation of muscular Hb/Mb, and the opening of the ATP-dependent K+ channels which can enhance the level of performance [3–5,8,9,12]. There are many different methods related to the point, timing of application and the duration of ischemia as part of physical exercise: pre-conditioning (ischemia induced only before a training session), continuous (ischemia induced during the exercise and rest periods), and intermittent (ischemia induced during exercise, not during rest periods) [13–15] The main differences between those methods are related to the point when ischemia is induced [16]. Recently, another method of using ischemia as part of resistance exercises has been introduced called ischemic intra-conditioning [14]. During ischemic intra-conditioning, the cuffs are applied only during each rest period between sets of resistance exercise and released before start effort [14]. Wilk et al. [14] showed an increase in bar velocity and power output during the bench press exercise (5 sets, 3 repetitions, 60%1RM; 5-minute rest periods) when ischemia (80% full arterial occlusion pressure (AOP) of the upper limb at rest) was applied before each set of resistance exercise. Since the ischemia used only during the rest periods increase explosive performance [17] it is possible that the positive effect may also apply to maximal strength as well as strength-endurance performance. Previous study showed that ischemic pre-conditioning increases maximal number of performed repetitions during resistance exercise for the lower and upper limbs [16,18–20]. Novaes et al. [20] showed that ischemic pre-conditioning applied 40 min before warm-up (4 cycles of 5-minute ischemia at 220mmHg) and completed 5 min before warm-up increased total training volume (sum of a total number of performed repetitions x load in bench press, leg press, lateral pulldown, hack machine squat, shoulder press, and Smith back squat) compared to control condition. Similar Marocolo et al. [19] showed that ischemic pre-conditioning (4 cycles of 5 minutes ischemia at 220mmHg) increased the maximal number of performed repetitions during leg extension in the first set and second set, but not in the third. Therefore generally, the ischemia used before resistance exercise increases the maximal number of performed repetitions, however currently, there are no studies assessing the impact of ischemic intra-conditioning on strength-endurance performance.

Since the ischemic pre-conditioning increase strength-endurance performance, it can be assumed that ischemic intra-conditioning could also cause positive effect on strength-endurance performance. Therefore, the present study aimed to evaluate the effects of ischemia used before the first set as well as during all rest periods between successive sets on maximal number of performed repetitions, time under tension and bar velocity during the bench press exercise.

It was hypothesized that the ischemic intra-conditioning would increase strength-endurance performance during the bench press exercise.

## Materials & methods

### Experimental approach to the problem

During the experimental protocols each participant performed two experimental sessions (one week apart) in a random order: a) when ischemia were used before each of exercise sets (ischemia condition); b) control, when ischemia were not used. In experimental protocol the subjects performed bench press exercise, 5 sets, at load 70%1RM and maximal number of repetitions in each. A 5-min rest-interval was used between each set. For ischemic session the cuffs pressures (10 cm wide cuff) was set to 80% AOP. Pneumatic cuffs were used on both arms: a) after warm-up before initial set; b) over all four brake interval between sets (for 4.5 minute). During the control sessions the cuffs were no used. The Bioethics Committee approved the experimental (number 02/2019).

### Subjects

Thirteen men (resistance trained) participated in the experimental (body mass = 87.2 ± 8.6 kg; age = 28.5 ± 7.1 years; bench press 1RM = 143.1 ± 20.7 kg; training experience = 11.0 ± 6.9 years; ratio strength = 1.60 ± 0.2 (1RM / body weight)). The following inclusion criteria were used: a) absence of muscular injuries b) bench press maximal strength (1RM) more than 150% body mass. Participants signed written informed consent to participate in the study and were allowed to withdraw from participation at any time.

### Procedures

**Familiarization session and one repetition maximum test (1RM).** The familiarization session was performed 2-weeks before the main experimental sessions. During the familiarization session each participant performed a warm-up close to their standard training habits. After warm-up, each participant performed bench press exercise (3 sets, around 60%1RM, maximal number of performed repetitions) with ischemia used only during rest periods. The ischemia was used for 4.5 minute and released thirty second before starting each set. A week before the main experimental sessions, the maximal strength test (1RM) in bench press was performed [14,21]. The 1RM test started with the standard warm-up (upper body). Next, was performed bench press at load 20%1RM (8 reps), 40%1RM (6 reps), and 60%1RM (3 reps). The first control load in bench press was set to presumed 70–80%1RM. Then the load increased by 5 to 10kg for each performed attempt. During each set one repetition was performed. The load increased until the maximum results were obtained.

### Main experimental trials

In counterbalanced and randomized order the participants performed two experimental sessions: (a) ischemia session, where the cuffs was used before each set of bench press (b) control session, where the subjects performed bench press exercise without ischemia. In both conditions the bench press exercise were performed (5 sets; load 70%1RM). In each set, the participant performed maximal possibility number of repetitions, with maximal speed movement in both phases of movement [22]. A 5-minutes rest periods between each sets was used. Velocity of the bar was recorded by Tendo Analyzer (Slovakia) [23]. The bar peak velocity (PV) values was received from the best repetition in each separately set. The bar mean velocity values was received from the all repetitions performed in each separately set [24]. The number of

performed repetitions as well as time under tension was obtained from the recorded videos. In order to ensure the reliability of manual video data collection, three persons made the data analysis.

### Ischemia

The cuffs used to induce ischemia (Smart Tools, USA; 10 cm wide) were applied in the most proximal area of both arms. The cuffs was applied 5 minutes before every performed set. The ischemia was induced for 4.5 minutes and additional 30s was intended to put on or take off the cuffs (inflate the cuffs—20s.; deflate the cuffs– 10s.). The cuffs have been completely removed when performing the bench press. The cuff pressure was approximately 80% AOP (114.0 ± 10.7 mmHg). To determine the individual pressure value, we used a standardized procedure described elsewhere [13,14].

### Statistical analysis

For statistical analyses was used Statistica 9.1. The Shapiro-Wilk test was used to verify normality, homogeneity of sample group. Statistical differences between ischemic and control conditions independently for mean velocity (MV), peak velocity (PV), time under tension and for number of performed repetitions were analyzed by two-way repeated ANOVA [(ischemic session vs. control session) × 5-sets bench press]. The partial eta squared were used to determined effect sizes (ES). Partial eta squared values were classified 0.01–0.059 as small, 0.06–0.137 as moderate, and >0.137 as large. Post hoc comparisons using the Tukey's test were conducted to locate the differences between mean values, when a main effect or an interaction was found. For pairwise comparisons, ESs were determined by Cohen's d which was characterized $d<0.2$ as trivial; d between 0.49 and 0.20 as small; d between 0.8 and 0.5 as moderate; $d > 0.8$ as large. The $p < 0.05$ was determined as statistical significance.

### Results

The results showed significant statistically interaction effect in time under tension ($p = 0.022$; $\eta^2 = 0.20$). Further, there was lack of significant statistically interaction effect for PV ($p = 0.28$; $\eta^2 = 0.10$) MV ($p = 0.38$; $\eta^2 = 0.08$), and for number of performed repetitions ($p = 0.28$; $\eta^2 = 0.09$).

The ANOVA also showed a statistically significant main effect of condition for time under tension ($p = 0.04$; $\eta^2 = 0.31$; 22.3 s vs. 23.7 s, respectively for ischemia and control conditions). Further there was no main effect of condition for number of performed repetitions ($p = 0.55$; $\eta^2 = 0.03$), for PV ($p = 0.90$; $\eta^2 < 0.01$) and for MV ($p = 0.26$; $\eta^2 = 0.10$).

The test Tukeya analysis for interaction showed significantly shorter time under tension for condition under ischemia compared to control group in first set ($p < 0.01$; 28.0 s vs. 31.4 s; Table 1). The test Tukeya analysis for main effect showed significant shorter time under tension for ischemia compared with the control condition ($p = 0.04$; 22.3 s vs. 23.7 s).

### Discussion

The result of this study showed that ischemic intra-conditioning did not increase the strength-endurance capabilities when the bench press at 70%1RM was performed. The result of the presented study shows lack of significant differences in a maximal number of performed repetitions, as well as a lack of differences in bar velocity between ischemia and control condition, when bench press exercise was performed to muscle failure. However, what is particularly important despite that was not observed differences in a maximal number of performed

**Table 1. Differences in performance variables during control and ischemia conditions.**

| Condition | Set 1 (95%CI) | Set 2 (95%CI) | Set 3 (95%CI) | Set 4 (95%CI) | Set 5 (95%CI) |
|---|---|---|---|---|---|
| Peak Bar Velocity [m/s] | | | | | |
| Control | 0.73 ± 0.07 (0.69 to 0.78) | 0.69 ± 0.08 (0.64 to 0.74) | 0.66 ± 0.09 (0.61 to 0.71) | 0.64 ± 0.10 (0.58 to 0.70) | 0.61 ± 0.10 (0.55 to 0.67) |
| Ischemia | 0.70 ± 0.07 (0.66 to 0.75) | 0.68 ± 0.09 (0.63 to 0.74) | 0.66 ± 0.10 (0.60 to 0.72) | 0.65 ± 0.12 (0.58 to 0.72) | 0.63 ± 0.11 (0.56 to 0.70) |
| ES | 0.43 | 0.12 | 0.00 | 0.09 | 0.19 |
| Mean Bar Velocity [m/s] | | | | | |
| Control | 0.56 ± 0.05 (0.53 to 0.59) | 0.52 ± 0.04 (0.50 to 0.55) | 0.52 ± 0.06 (0.48 to 0.56) | 0.46 ± 0.07 (0.42 to 0.50) | 0.47 ± 0.06 (0.44 to 0.51) |
| Ischemia | 0.54 ± 0.05 (0.51 to 0.58) | 0.54 ± 0.07 (0.50 to 0.58) | 0.51 ± 0.08 (0.46 to 0.56) | 0.44 ± 0.10 (0.38 to 0.50) | 0.43 ± 0.05 (0.40 to 0.46) |
| ES | 0.40 | 0.35 | 0.14 | 0.09 | 0.72 |
| Number of performed repetitions [n] | | | | | |
| Control | 16.8 ± 3.4 (14.8 to 18.9) | 13.7 ± 1.4 (12.8 to 14.6) | 11.9 ± 1.0 (11.3 to 12.6) | 9.8 ± 1.2 (9.1 to 10.5) | 9.4 ± 1.3 (8.6 to 10.1) |
| Ischemia | 16.5 ± 2.0 (15.3 to 17.8) | 14.3 ± 2.3 (12.9 to 15.7) | 12.2 ± 1.6 (11.2 to 13.2) | 10.3 ± 1.3 (9.6 to 11.1) | 9.3 ± 1.4 (8.4 to 10.2) |
| ES | 0.11 | 0.32 | 0.22 | 0.40 | 0.07 |
| Time Under Tension [s] | | | | | |
| Control | 31.4 ± 2.9 (29.7 to 33.2) | 24.8 ± 2.8 (23.1 to 26.4) | 23.4 ± 2.3 (22.0 to 24.8) | 20.4 ± 3.3 (18.5 to 22.4) | 18.6 ± 2.3 (17.2 to 20.0) |
| Ischemia | *28.0 ± 4.0 (25.6 to 30.5) | 24.4 ± 3.8 (22.2 to 26.7) | 21.6 ± 2.9 (19.9 to 23.3) | 19.2 ± 2.5 (17.7 ± 20.7) | 18.3 ± 2.8 (16.6 to 20.0) |
| ES | 0.97 | 0.12 | 0.69 | 0.41 | 0.12 |

All data are presented as mean with standard deviation [SD]; CI = confidence interval; ES = Cohen's d;

*Statistically significant differences in comparison with the corresponding value in control condition p < 0.05.

repetitions between conditions, the result of the presented study shows a decrease in time under tension for ischemia compared to control condition.

Currently, there is only one available study that investigated the impact of ischemic intra-conditioning (restriction used only during the rest periods between sets) in resistance exercise [14]. These authors showed that ischemic intra-conditioning increased bar velocity and power output during the bench press performed with a load of 60% of 1RM (5 sets of 3 repetitions with 5-minute rest between sets). However, this is the first study that investigated the effects of ischemic intra-conditioning in strength-endurance performance. The result of present study did not show differences in number of performed repetitions, and in bar velocity (both PV and MV) between ischemia and control condition during the five sets of bench press exercise performed to failure. In the study by Wilk et al. [14] the experimental procedure contains a lower number of repetitions (only 3 reps in each set) lasting approximately 3–5 s per set while in present study each set was performed to muscle failure and lasted 18–32 s. It seems that the duration of exercise or fact that the successive sets to failure are performed may determine the acute ischemic intra-conditioning effect, hence the differences in outcomes between our result and study Wilk et al. [14]. Therefore, the lack of changes in strength- endurance performance for ischemic condition compared to study Wilk et al. [14] may be related to the longer duration of the effort. Ischemia before effort increase exercise physical performance as well as stimulates physiological responses such as improvement of metabolic efficiency (attenuating ATP

depletion), also glycogen depletion and the lactate production [2–11]. Further ischemia causes changes in systemic VO2, in the deoxygenation of muscular Hb/Mb, and the opening of the ATP-dependent K+ channels which can enhance the level of performance [3–5,8,9,12]. However, these physiological responses may have a positive effect on physical performance, but when the effort after the ischemia is not performed several times to failure. It seems that the duration of exercise or fact that the successive sets to failure are performed may determine the acute ischemic intra-conditioning effect, hence the differences in outcomes between our result and study Wilk et al. [14].

Nonetheless, it should be noted that although the present study did not show differences in maximal number of performed repetitions between conditions, there was observed decrease in time under tension for ischemia condition. The shorter time under tension without changes in the number of performed repetitions can be related with the increase of velocity in eccentric movement as it was recoded in study Wilk et al. [24]. However, in presented study the eccentric velocity was not measured. Therefore, based on result of time under tension it can be concluded that ischemic intra-conditioning decrease maximal time of effort during resistance exercise. However, performing repetitions to exhaustion in each set with additional ischemia applied in rest periods would be expected to cause significant reductions in exercise capacity [25] not only in time under tension but also reductions of explosive performance however such reductions were not observed. The effect of maintaining a certain amount of power output during progressive fatigue following ischemic intra-conditioning was previously observed also in a study by Trybulski et al. [26] showed that ischemia applied during the rest period between sets of Kaiser Squats prevented progressive fatigue compared to the control condition which could also take place in the our study. Maintaining peak and mean concentric velocity during ischemia condition, may be related to the fact that ischemic intra-conditioning compensates or limits the negative effects of arising fatigue. The muscles blood flow previously subjected to ischemic pre-conditioning, become more resistant to arising acute effort fatigue and its potential adverse effects on performance [27], therefore it can be assumed that a similar effect will occur for both ischemic pre and intra-conditioning used during one training protocol. The previous studies showed that the ischemia used only as pre-conditioning increased the strength-endurance performance [16,18–20,28]. Novaes et al. [20] showed that total training volume (the number of performed repetitions) increased following the ischemic pre-conditioning. A similar result was observed in a study by Marocolo et al. [19]. Ischemic pre-conditioning increase physical performance and stimulates acute responses such as improvement of metabolic efficiency by attenuating ATP depletion, as well as glycogen depletion and post exercise lactate production [2–11]. Furthermore, ischemic as pre-conditioning improve blood flow in skeletal muscles, by inducing vasodilation [29], improving functional sympatholysis [18] and preserving microvascular endothelium function during stress [30–33]. The similar effect can be observed when ischemic intra-conditioning is used. Although some studies have showed positive effects of pre-conditioning ischemia on physiological responses and performance, not all studies such report beneficial effects. Marocolo et al. [34] did not observe changes in performance following ischemic pre-condition, while Paixao et al. [35] showed negative effect of ischemic pre-condition. Therefore the effect of ischemia on performance are highly contentious [15] and its effectiveness may depend on the methodology of using ischemia (time when ischemia is applied, duration of ischemia, number of cycles, pressure of cuffs). However, even if the ischemia used during rest periods doesn't increase performance it should be noted that this ischemic intra-conditioning does not decrease a maximal number of performed repetitions, as well as peak and mean bar velocity.

An important factor influencing the efficiency of different schemes of ischemia are number of ischemic cycles during training sessions and the duration of a single ischemia cycle as well

as pressure of cuffs. Marocolo et al. [19] showed that ischemic pre-conditioning (4 cycles of 5 minutes ischemia at 220mmHg) increased the maximal number of performed repetitions during leg extension in the first set and second set, but not in the third. Contrary, Cocking et al. [36] showed that five cycles of pre-conditioning ischemia cycles (5 cycles for 5 min, cuffs pressure 220mm/Hg) have not promoted enhancements in exercise performance, similar to our study where five ischemic intra-conditioning cycles (5 cycles for 4.5 min, cuffs pressure 114mmHg) do not induce enhancements in strength-endurance performance. Therefore, its seems that such ischemia protocols is insufficient or too burdensome to induce positive responses in physical performance. Furthermore, in the presented study also single pre-conditioning ischemia before first set of bench press exercise did not increase strength-endurance performance and explosive performance for first set. Therefore, both the repetitive 4.5-min ischemia did not cause any changes in the number of performed repetitions, but also single 4.5-min pre-conditioning ischemia was not optimal to cause positive change in the 1<sup>st</sup> set of exercise.

Although the results of the present study expands knowledge and training clues, there are some study limitations that should be addressed. In presented study the physiological or metabolic responses was not determined what is main limitation which should be addressed. Furthermore, the variations of pressure of ischemia as well as different time of ischemia may also have a significant impact on endurance-performance, which require further studies.

## Conclusions

This study indicate that ischemic intra-conditioning does not increase strength-endurance performance as well as bar velocity in bench press exercise performed to muscle failure. Therefore, ischemic intra-conditioning is not effective method to increase strength-endurance performance of upper body. However, even if the ischemia used during rest periods do not increase performance it should be noted that this ischemic intra-conditioning does not decrease maximal number of performed repetitions, as well as peak and mean bar velocity. Therefore, the maintaining volume of effort and peak and mean concentric velocity during ischemia condition, while possibly increasing physiological responses, may increase acute responses which may indirectly affect chronic muscle adaptation.

## Supporting information

**S1 Raw data.**
(XLSX)

## Author Contributions

**Conceptualization:** Michal Krzysztofik, Michal Wilk.

**Formal analysis:** Michal Wilk.

**Funding acquisition:** Piotr Makar.

**Investigation:** Robert Trybulski, Marta Bichowska, Rafal Piwowar, Anna Pisz.

**Methodology:** Robert Trybulski, Rafal Piwowar, Aleksandra Filip-Stachnik.

**Project administration:** Robert Trybulski, Rafal Piwowar, Anna Pisz.

**Resources:** Marta Bichowska, Anna Pisz, Aleksandra Filip-Stachnik, Krzysztof Fostiak.

**Software:** Krzysztof Fostiak, Piotr Makar.

**Supervision:** Michal Wilk.

**Writing – original draft:** Robert Trybulski, Rafal Piwowar, Anna Pisz.

**Writing – review & editing:** Michal Krzysztofik, Michal Wilk.

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
