## [Decision Letter · Decision Letter 0]

30 Mar 2022

PONE-D-21-36446The effects of ischemia during rest intervals on strength endurance performancePLOS ONE

Dear Dr. Wilk,

Thank you for submitting your manuscript to PLOS ONE. After careful consideration, we feel that it has merit but does not fully meet PLOS ONE’s publication criteria as it currently stands. Therefore, we invite you to submit a revised version of the manuscript that addresses the points raised during the review process.

Sorry for the delay but it was very hard to find reviewers for this manuscript. As we have only one review report, I would like to invite you to respond it and suggesting me other potential reviewers during the next round of revisions to achieve the minimum number of reviewers required for peer review (2).

We look forward to receiving your revised manuscript.

Kind regards,

Daniel Boullosa

Academic Editor

PLOS ONE

Journal Requirements:

The  study  was  supported  and  funded  by  the  statutory  research  of  the  Jerzy  Kukuczka  Academy  of  Physical  Education  in  Katowice,  Poland. 

4. We note you have included a table to which you do not refer in the text of your manuscript. Please ensure that you refer to Table 1 in your text; if accepted, production will need this reference to link the reader to the Table.

6. Thank you for submitting the above manuscript to PLOS ONE. During our internal evaluation of the manuscript, we found significant text overlap between your submission and the following previously published works, some of which you are an author.

- https://www.frontiersin.org/articles/10.3389/fphys.2021.715096/full

Please revise the manuscript to rephrase the duplicated text, cite your sources, and provide details as to how the current manuscript advances on previous work. Please note that further consideration is dependent on the submission of a manuscript that addresses these concerns about the overlap in text with published work.

Reviewers' comments:

Reviewer's Responses to Questions

**Comments to the Author**

1. Is the manuscript technically sound, and do the data support the conclusions?

Reviewer #1: Partly

2. Has the statistical analysis been performed appropriately and rigorously? 

Reviewer #1: Yes

3. Have the authors made all data underlying the findings in their manuscript fully available?

Reviewer #1: Yes

4. Is the manuscript presented in an intelligible fashion and written in standard English?

Reviewer #1: Yes

5. Review Comments to the Author

Reviewer #1: This is an interesting study where authors investigate the effects of ischemia applied before the first set and between sets of resistance exercise (bench press) performed to failure on number of repetitions, time under tension and bar velocity. The authors conclude that ischemia applied between-sets does not increase strength-endurance nor bar velocity during bench press exercise performed to muscle failure.

I hope that my comments be helpful to improve the readability of the manuscript.

General comments

The manuscript is well written and easy to follow. Although I am not a native English speaker, I suggest some minor adjustments in writing. The objective is clear and the results are important to the practitioner. Some caution must be exercised when trying to speculate the results from acute study to long term adaptations. I believe the discussion section needs some work to improve the interpretation of the results.

Specific comments.

Introduction

I would not say that ischemia is the same as blood flow restriction, as ischemia may result from the external compression, but it is not the method.

L 64-65 - What do the authors mean by “The differences between those methods are related to the point when ischemia is applied”. This sentence refers to using restriction continuous, intermittent or pre-conditioning.

Also, when BFR is used during resistance training, the aim is to partially blood inflow and fully restrict blood outflow in the exercising muscles. This is not the aim of the ischemic pre-conditioning. So I suggest that this paragraph be rewritten.

L62- replace combine with combined

L67 – I suggest that “ischemic pre-conditioning” be used instead of “ischemia pre-conditioning”

L68 – please use the same term – ischemia intra-conditioning (or ischemic, if you choose to replace), but do not use “intra-conditioning ischemia”. The same comment applies to other places in the manuscript where “intra-conditioning ischemia” is used.

L75 – I suggest that “… it can be assumed…” be replaced with “… it is possible that…”.

L77 – performed

L79 – ischemic pre-conditioning. I suggest that the authors include when the ischemic pre-conditioning was applied. It is possible that some readers are not familiar with this strategy.

L83 -replace “a” with “the”

L86 -replace “a” with “the”

L89 – Please include what was assessed as physical performance?

L91-93 – In this study, ischemia was applied not only between sets, but also previous to the beginning of the training sets.

L94-95 – Please rewrite the sentence starting with “It was hypothesized…”

Methods

This section is easy to follow.

I wonder why bar velocity and time under tension were assessed if the objective was to investigate the effects of ischemic intra-conditioning in strength endurance performance. It is not in the introduction; they are not markers of strength endurance and is briefly discussed in the discussion section.

L 158- coefficient

L182-184 – please, rewrite. It is confusing. What tests were used?

Results

Table 1 – specify what is present in parentheses. I believe it is the CI. Specify that the ES presented is Cohen’s d.

Discussion

For me, this is the critical part of this manuscript, and authors need to rewrite some of the information to improve flow and readability. For example, in the 2nd paragraph, authors mention the results of bar velocity and power and contrast them to the findings of strength endurance. These variables are very different.

Also, I suggest that authors organize the discussion according to the variables they investigate, and then try to make a final statement gathering all the discussion presented.

L228 - replace author’s with authors’

L 239 – 241 – There is a comparison with the study by Wilk who observed that ischemic intra-conditioning improved bar velocity is a training session consisting of 5 sets with 3 reps at 60% 1RM (maximal number of repetitions was not assessed). Why authors try to discuss the different set durations if the first study did not assess strength endurance?

The important question that has to discussed is why was not there difference between experimental and control condition in the present study?

When authors bring the study of Wilk I imagined they would discuss the lack of effects in bar velocity. This is an interesting discussion. The highest peak velocity and mean velocity should have presented the same pattern in this study and Wilk’s, why it did not happen?

Time under tension decreased and number of repetitions was not affected, how do you explain that?

L47-L251 – This is important information and should be discussed further

L252- I am not sure that physiological responses should be discussed here, as you did not assess them. But Why would physiological responses increase during ischemia condition?

L257 – it is not just intra-conditioning that can have affected the results, but also the pre-conditioning.

L277 – replace do with does

L280- Please discuss the pressure applied

L285 – delete was

L286- delete the

L310 – Present the limitation in a different paragraph.

L324 – 327 – This is speculation and the authors have no results to support it. It should not be in the conclusion.

References

Please correct the references as there are no journal names.

6. PLOS authors have the option to publish the peer review history of their article (what does this mean?). If published, this will include your full peer review and any attached files.

Reviewer #1: **Yes: **Renato Barroso

---

## [Author Response · Author response to Decision Letter 0]

7 Aug 2022

Reviewer #1: This is an interesting study where authors investigate the effects of ischemia applied before the first set and between sets of resistance exercise (bench press) performed to failure on number of repetitions, time under tension and bar velocity. The authors conclude that ischemia applied between-sets does not increase strength-endurance nor bar velocity during bench press exercise performed to muscle failure.

I hope that my comments be helpful to improve the readability of the manuscript.

Reply - We are grateful for the commitment and very valuable comments that helped us to improve the quality of the manuscript. We hope that the revised manuscript adequately addresses all the raised issues.

General comments

The manuscript is well written and easy to follow. Although I am not a native English speaker, I suggest some minor adjustments in writing. The objective is clear and the results are important to the practitioner. Some caution must be exercised when trying to speculate the results from acute study to long term adaptations. I believe the discussion section needs some work to improve the interpretation of the results.

Specific comments.

Introduction

I would not say that ischemia is the same as blood flow restriction, as ischemia may result from the external compression, but it is not the method.

Reply – yes we agree with the reviewer that ischemia may result from external compression, that's why we made such a change to: “The ischemia-induced by the external compression or by the BFR cuffs” 

L 57-58

L 64-65 - What do the authors mean by “The differences between those methods are related to the point when ischemia is applied”. This sentence refers to using restriction continuous, intermittent or pre-conditioning.

Also, when BFR is used during resistance training, the aim is to partially blood inflow and fully restrict blood outflow in the exercising muscles. This is not the aim of the ischemic pre-conditioning. So I suggest that this paragraph be rewritten.

Reply - I have corrected the indicated sentence L60-64

L62- replace combine with combined

Reply – the change has been made 

L67 – I suggest that “ischemic pre-conditioning” be used instead of “ischemia pre-conditioning”

Reply – the change has been mad

L68 – please use the same term – ischemia intra-conditioning (or ischemic, if you choose to replace), but do not use “intra-conditioning ischemia”. The same comment applies to other places in the manuscript where “intra-conditioning ischemia” is used.

Reply - Thank you, I have standardized the term throughout

L75 – I suggest that “… it can be assumed…” be replaced with “… it is possible that…”.

Reply – the change has been made

L77 – performed

Reply – the change has been made

L79 – ischemic pre-conditioning. I suggest that the authors include when the ischemic pre-conditioning was applied. It is possible that some readers are not familiar with this strategy.

Reply – we added such information about when the ischemia was applied L77- 78

L83 -replace “a” with “the”

Reply – the change has been made

L86 -replace “a” with “the”

Reply – the change has been made

L89 – Please include what was assessed as physical performance?

Reply – the change has been made L79-81

L91-93 – In this study, ischemia was applied not only between sets, but also previous to the beginning of the training sets.

Reply - I added information on application of ischemia also before the first set L91

L94-95 – Please rewrite the sentence starting with “It was hypothesized…”

Reply – the sentence was rewritten L 93-95

Methods

This section is easy to follow.

I wonder why bar velocity and time under tension were assessed if the objective was to investigate the effects of ischemic intra-conditioning in strength endurance performance. It is not in the introduction; they are not markers of strength endurance and is briefly discussed in the discussion section.

Reply – time under tension is increasingly used as an indicator of exercise volume, more reliable than the number of repetitions, and the value of the maximal exercise volume performed is an important aspect in assessing strength-endurance performance. Further also changes in bar velocity may indicate the level of increasing fatigue. Therefore, both parameters can be important parameters in the assessment of the impact of BFR on the level of strength-endurance performance. 

L 158- coefficient

Reply - word was corrected

L182-184 – please, rewrite. It is confusing. What tests were used?

Reply – sentence was corrected 

Results

Table 1 – specify what is present in parentheses. I believe it is the CI. Specify that the ES presented is Cohen’s d.

Reply – yes it is CI. The table was corrected 

Discussion

For me, this is the critical part of this manuscript, and authors need to rewrite some of the information to improve flow and readability. For example, in the 2nd paragraph, authors mention the results of bar velocity and power and contrast them to the findings of strength endurance. These variables are very different.

Also, I suggest that authors organize the discussion according to the variables they investigate, and then try to make a final statement gathering all the discussion presented.

Reply – The main problem with preparing the discussion is that there are no other studies about intra-conditioning and strength-endurance performance to which we could compare our results, so we used the results from other ischemia methods (ischemic pre-conditioning) or other measuring and other variables. However, as suggested by the reviewer, we have made changes to the discussion structure

L228 - replace author’s with authors’

Reply – done 

L 239 – 241 – There is a comparison with the study by Wilk who observed that ischemic intra-conditioning improved bar velocity is a training session consisting of 5 sets with 3 reps at 60% 1RM (maximal number of repetitions was not assessed). Why authors try to discuss the different set durations if the first study did not assess strength endurance?

Reply - we decided to make this comparison because there are no other studies that have used ischemia only during rest intervals

The important question that has to discussed is why was not there difference between experimental and control condition in the present study?

When authors bring the study of Wilk I imagined they would discuss the lack of effects in bar velocity. This is an interesting discussion. The highest peak velocity and mean velocity should have presented the same pattern in this study and Wilk’s, why it did not happen?

Time under tension decreased and number of repetitions was not affected, how do you explain that?

Reply - as suggested by the reviewer, we have added appropriate explanations L242-251

L47-L251 – This is important information and should be discussed further

Reply - the discussion in this regard has been expanded L255-259

L252- I am not sure that physiological responses should be discussed here, as you did not assess them. But Why would physiological responses increase during ischemia condition?

Reply - yes, the reviewer is right, so we have removed the term potential increases of physiological responses during ischemia condition. 

L257 – it is not just intra-conditioning that can have affected the results, but also the pre-conditioning.

Reply – yes, we made appropriate changes 

L277 – replace do with does

Reply – done 

L280- Please discuss the pressure applied 

Reply - Ghosh et al. does not provide such information therefore we decide to delete this ref. In other refs. we added relevant information L287-303 

L285 – delete was

Reply – done 

L286- delete the

Reply – done 

L310 – Present the limitation in a different paragraph.

Reply – done 

L324 – 327 – This is speculation and the authors have no results to support it. It should not be in the conclusion.

Reply - I agree with the reviewer's opinion, therefore I deleted the indicated sentence

References

Please correct the references as there are no journal names.

Reply - In fact, I haven't noticed such a big mistake. Thank you for paying attention

---

## [Decision Letter · Decision Letter 1]

27 Sep 2022

PONE-D-21-36446R1The effects of ischemia during rest intervals on strength endurance performancePLOS ONE

Dear Dr. Wilk,

Thank you for submitting your manuscript to PLOS ONE. After careful consideration, we feel that it has merit but does not fully meet PLOS ONE’s publication criteria as it currently stands. Therefore, we invite you to submit a revised version of the manuscript that addresses the points raised during the review process.

We look forward to receiving your revised manuscript.

Kind regards,

Daniel Boullosa

Academic Editor

PLOS ONE

Reviewers' comments:

Reviewer's Responses to Questions

**Comments to the Author**

1. If the authors have adequately addressed your comments raised in a previous round of review and you feel that this manuscript is now acceptable for publication, you may indicate that here to bypass the “Comments to the Author” section, enter your conflict of interest statement in the “Confidential to Editor” section, and submit your "Accept" recommendation.

Reviewer #1: (No Response)

2. Is the manuscript technically sound, and do the data support the conclusions?

Reviewer #1: No

3. Has the statistical analysis been performed appropriately and rigorously? 

Reviewer #1: Yes

4. Have the authors made all data underlying the findings in their manuscript fully available?

Reviewer #1: Yes

5. Is the manuscript presented in an intelligible fashion and written in standard English?

Reviewer #1: No

6. Review Comments to the Author

Reviewer #1: Thanks for the opportunity to review the revised version of the manuscript.

As I mentioned in the previous review, Ischemia is not the same as blood flow restriction.

The first sentence of the introduction states that. “Ischemia also referred blood flow restriction (BFR)…”. Ischemia is the “deficient supply of blood to a body part (such as the heart or brain) that is due to obstruction of the inflow of arterial blood” (Merriam-Webster dictionary. Thus, I suggest the term BFR be used with caution and not as synonymous of ischemia.

Methods, Results and Discussion sections are difficult to understand.

In L59-60 – It is stated that there are different methods of applying BFR. I would expect that authors mentioned “how” the restriction was applied and not “when”. In this part of the manuscript, I believe that the main point should be on the ischemia and “when” it is induced, which brings the problem that was investigated: Intra-conditioning ischemia.

Maybe some adjustment in the terminology used is necessary. The use of ischemia before the exercise or before transplantation (as it was originally used) has been called Ischemic pre-conditioning. I suggest that authors use ischemic intra-conditioning, instead of ischemia intra-conditioning. Ischemia is a noun while ischemic is an adjective, which is the case as “ischemic” is characterizing the “pre-conditioning”.

However, if I had to hypothesize something, I would say that ischemic intra-conditioning would impair strength and endurance performance, as it reduces the supply of oxygen to the working muscles, which is necessary to the recovery of substrates and removal of metabolites.

L73 – replace “Previously” with “previous”

L87 – conditioning instead of condition

L88 – delete but

L90 – “…before THE first set…”. Delete “the” before “… all rest…”

L102 – A 5-min rest-interval was used between each set.

L103 – ischemic instead of ischemia.

L104 – Pneumatic cuffs were used on both arms…

L105 – replace brake with interval. “Brake” refers to something used to slow down or stop movement.

L109 – replace take part with participated

L112 – replace “lack” with “absence”.

Avoid starting a sentence with a number.

Please rewrite “Procedures” section

How time under tension was calculated?

L152 – delete “To” (last word in this line)

L154-155 – please rewrite the sentence starting with . The statistical differences…

What variables were analyzed? And how?

L162 – What does li mean?

If there was an effect, please present what was the direction of the effect. For example, PV was higher in Condition I compared to condition II (p….).

L170 – Tukeya?

L172 – Tukeya?

In the table – How ES was calculated? It seems that it refers to Cohen’s d, but in the statistical analysis section, ES refers to eta squared.

Discussion section

L185-186 – I would leave the last sentence of this paragraph to the conclusion.

I suggest that discussion section be rewritten to improve the understanding. I have made a suggestion in the second paragraph of the discussion.

Currently, there is only one available study that investigated the impact of ischemic intra-conditioning (restriction used only during the rest periods between sets) in resistance exercise (3). These authors showed that ischemic intra-conditioning increased bar velocity and power output during the bench press performed with a load of 60% of 1RM (5 sets of 3 repetitions with 5-minute rest between sets). However, this is the first study that investigated the effects of ischemic intra-conditioning in strength-endurance performance. The result of present study did not show differences in number of performed repetitions, and in bar velocity (both PV and MV) between ischemia and control condition during the five sets of bench press exercise performed to failure. In the study by Wilk et al. (3) the experimental procedure contains a lower number of repetitions (only 3 reps in each set) lasting approximately 3–5 s per set while in present study each set was performed to muscle failure and lasted 18-32 s. It seems that the duration of exercise or fact that the successive sets to failure are performed may determine the acute ischemia intra-conditioning effect, hence the differences in outcomes between our result and study Wilk et al. (3). Therefore, the lack of changes in strength- endurance performance for ischemic condition compared to study Wilk et al. (3) may be related to the longer duration of the effort.

I am not sure how the physiological mechanisms can help in the discussion. For the mechanisms presents. Number of repetitions should have been improved, which was not observed. I wonder if there were other mechanisms involved.

L215- present instead of presented. Did not instead of didn’t

L218 – IN THE number of repetitions….

L220 – present. Replace measurement with measured.

L220-224 – what is the rationale of suggesting that time under tension is a better indicator of training volume than repetition, and why is it being mentioned here? Are the authors suggesting that training without ischemic intra-conditioning would result in smaller changes in strength and muscle size even with the same number of repetitions performed? What is the rationale? Be careful when extrapolating results from acute studies to long term adaptations. Maybe the following reference can help: Carvalho L, Concon V, Meloni M, De Souza EO, Barroso R. Effects of resistance training combined with ischemic preconditioning on muscle size and strength in resistance-trained individuals. The Journal of sports medicine and physical fitness. 2020 Nov;60(11):1431-6.

As I mentioned before, if I had to guess, I would say that ischemic condition would impair strength and endurance performance. I enjoyed the discussion in L230-259.

I am not sure that the discussion L260-281 is necessary. Also, it seems contradictory as Wilk showed that 4 bouts of 5min of ischemia induced important changes in power and bar velocity.

Discussion starting in L282 is speculative. It is not even known if acute “physiological, metabolite and hormonal responses” after ischemic condition are different.

7. PLOS authors have the option to publish the peer review history of their article (what does this mean?). If published, this will include your full peer review and any attached files.

Reviewer #1: No

---

## [Author Response · Author response to Decision Letter 1]

6 Nov 2022

Reviewer #1: Thanks for the opportunity to review the revised version of the manuscript.

Reply – Dear Reviewer. 

Once again thank you very much for your great commitment in reviewing the article and valuable comments. I made all necessary corrections and I believe that the article gained a lot of scientific value

As I mentioned in the previous review, Ischemia is not the same as blood flow restriction.

The first sentence of the introduction states that. “Ischemia also referred blood flow restriction (BFR)…”. Ischemia is the “deficient supply of blood to a body part (such as the heart or brain) that is due to obstruction of the inflow of arterial blood” (Merriam-Webster dictionary. Thus, I suggest the term BFR be used with caution and not as synonymous of ischemia.

Reply - Thank you for pointing out the differences in BFR and ischemia. In order to make the content of the article clearer, we have limited the use of the term BFR (L56-58; 65-69)

Methods, Results and Discussion sections are difficult to understand.

Reply - in line with the comments, I have revised these sections for the better of clarity

In L59-60 – It is stated that there are different methods of applying BFR. I would expect that authors mentioned “how” the restriction was applied and not “when”. In this part of the manuscript, I believe that the main point should be on the ischemia and “when” it is induced, which brings the problem that was investigated: Intra-conditioning ischemia.

Reply – I made the appropriate corrections (L59-65)

Maybe some adjustment in the terminology used is necessary. The use of ischemia before the exercise or before transplantation (as it was originally used) has been called Ischemic pre-conditioning. I suggest that authors use ischemic intra-conditioning, instead of ischemia intra-conditioning. Ischemia is a noun while ischemic is an adjective, which is the case as “ischemic” is characterizing the “pre-conditioning”.

Reply – I made the appropriate corrections

However, if I had to hypothesize something, I would say that ischemic intra-conditioning would impair strength and endurance performance, as it reduces the supply of oxygen to the working muscles, which is necessary to the recovery of substrates and removal of metabolites.

Reply - Initially, when I started researching the impact of ischemic intra-conditioning on strength performance, I also assumed that theoretically, it should decrease strength capabilities. However, several previous studies especially in the field of power output showed on the contrary that ischemic intra-conditioning caused an improvement in power performance. Therefore this topic seems to be very interesting

L73 – replace “Previously” with “previous”

Reply – Change has been made L80

L87 – conditioning instead of condition

Reply – done L93

L88 – delete but

Reply - done 

L90 – “…before THE first set…”. Delete “the” before “… all rest…”

Reply – done 

L102 – A 5-min rest-interval was used between each set.

Reply – the sentence was changed L108

L103 – ischemic instead of ischemia.

Reply – done L109

L104 – Pneumatic cuffs were used on both arms…

Reply – change has been made L110

L105 – replace brake with interval. “Brake” refers to something used to slow down or stop movement.

Reply – this sentence was rewritten L111

L109 – replace take part with participated

Reply – done L115

L112 – replace “lack” with “absence”.

Reply – done L118

Avoid starting a sentence with a number.

Reply - I changed the number to the word L129

Please rewrite “Procedures” section

Reply - I have made corrections in procedure section

How time under tension was calculated?

Reply – I have added information on how the TUT data was collected L149-151

L152 – delete “To” (last word in this line)

Reply – done 

L154-155 – please rewrite the sentence starting with . The statistical differences…

What variables were analyzed? And how?

Reply - I have made the appropriate corrections and added the information about which variables were analyzed L165-175

L162 – What does li mean?

Reply - I corrected this error

If there was an effect, please present what was the direction of the effect. For example, PV was higher in Condition I compared to condition II (p….).

Reply - in the case of significant differences, I added information about the conditions between which L165-175

L170 – Tukeya?

Reply – yes, test Tukeya

L172 – Tukeya?

Reply - yes, test Tukeya

In the table – How ES was calculated? It seems that it refers to Cohen’s d, but in the statistical analysis section, ES refers to eta squared.

Reply - I made the appropriate correction in the statistics section L165-175

Discussion section

L185-186 – I would leave the last sentence of this paragraph to the conclusion.

Reply - As suggested, I moved this sentence to conclusion 

I suggest that discussion section be rewritten to improve the understanding. I have made a suggestion in the second paragraph of the discussion.

Currently, there is only one available study that investigated the impact of ischemic intra-conditioning (restriction used only during the rest periods between sets) in resistance exercise (3). These authors showed that ischemic intra-conditioning increased bar velocity and power output during the bench press performed with a load of 60% of 1RM (5 sets of 3 repetitions with 5-minute rest between sets). However, this is the first study that investigated the effects of ischemic intra-conditioning in strength-endurance performance. The result of present study did not show differences in number of performed repetitions, and in bar velocity (both PV and MV) between ischemia and control condition during the five sets of bench press exercise performed to failure. In the study by Wilk et al. (3) the experimental procedure contains a lower number of repetitions (only 3 reps in each set) lasting approximately 3–5 s per set while in present study each set was performed to muscle failure and lasted 18-32 s. It seems that the duration of exercise or fact that the successive sets to failure are performed may determine the acute ischemia intra-conditioning effect, hence the differences in outcomes between our result and study Wilk et al. (3). Therefore, the lack of changes in strength- endurance performance for ischemic condition compared to study Wilk et al. (3) may be related to the longer duration of the effort.

Reply - thank you very much for the huge contribution to the proofreading, I agree that the current form of this paragraph is much better and clear. L204-220

I am not sure how the physiological mechanisms can help in the discussion. For the mechanisms presents. Number of repetitions should have been improved, which was not observed. I wonder if there were other mechanisms involved.

Reply - In line with my earlier comment, I have moved the potential physiological mechanisms to the introduction. In line with previous studies, pre-conditioning ischemia increased number of performed repetitions during leg extension, however our study did not show such effect. Therefore, what I pointed out in the introduction, the methodology of applying ischemia (when is used) may play a key role in assessing its impact ischemia on acute performance changes. (L270-272)

L215- present instead of presented. Did not instead of didn’t

Reply – done L232

L218 – IN THE number of repetitions….

Reply – done L235

L220 – present. Replace measurement with measured.

Reply – done L237

L220-224 – what is the rationale of suggesting that time under tension is a better indicator of training volume than repetition, and why is it being mentioned here? Are the authors suggesting that training without ischemic intra-conditioning would result in smaller changes in strength and muscle size even with the same number of repetitions performed? What is the rationale? Be careful when extrapolating results from acute studies to long term adaptations. Maybe the following reference can help: Carvalho L, Concon V, Meloni M, De Souza EO, Barroso R. Effects of resistance training combined with ischemic preconditioning on muscle size and strength in resistance-trained individuals. The Journal of sports medicine and physical fitness. 2020 Nov;60(11):1431-6.

Reply – yes, I agree that this paragraph was to speculative. Therefore I decide to delete this sentence. 

As I mentioned before, if I had to guess, I would say that ischemic condition would impair strength and endurance performance. I enjoyed the discussion in L230-259.

Reply - Initially, when I started researching the impact of ischemic intra-conditioning on strength performance, I also assumed that theoretically, it should decrease strength capabilities. However, several previous studies especially in the field of power output showed on the contrary that ischemic intra-conditioning caused an improvement in power performance. 

I am not sure that the discussion L260-281 is necessary. Also, it seems contradictory as Wilk showed that 4 bouts of 5min of ischemia induced important changes in power and bar velocity.

Reply - there are studies that indicate that the number of cycles of ischemia are important, also from a practical point of view it also seems to be useful information. I have kept the length of this section to a minimum

Discussion starting in L282 is speculative. It is not even known if acute “physiological, metabolite and hormonal responses” after ischemic condition are different.

Reply - I agree with the reviewer opinion and this paragraph has been removed

Thank you.

Best regards

Michal Wilk

---

## [Decision Letter · Decision Letter 2]

1 Dec 2022

PONE-D-21-36446R2The effects of ischemia during rest intervals on strength endurance performancePLOS ONE

Dear Dr. Wilk,

Thank you for submitting your manuscript to PLOS ONE. After careful consideration, we feel that it has merit but does not fully meet PLOS ONE’s publication criteria as it currently stands. Therefore, we invite you to submit a revised version of the manuscript that addresses the points raised during the review process.

Please, address the best you can the expert reviewers's concerns to avoid another round of revisions before acceptance. I agree with the reviewer that you should be precise with the language used. Please, communicate exactly what you did observe avoiding extrapolations. Personally, I don't like the concept of "strength endurance performance" as it is an oxymoron (i.e. nor it is strength nor it is endurance) but may accept it as it is your paradigm. However, I should alert you that this concept may confound the readers.

We look forward to receiving your revised manuscript.

Kind regards,

Daniel Boullosa

Academic Editor

PLOS ONE

Journal Requirements:

Reviewers' comments:

Reviewer's Responses to Questions

**Comments to the Author**

1. If the authors have adequately addressed your comments raised in a previous round of review and you feel that this manuscript is now acceptable for publication, you may indicate that here to bypass the “Comments to the Author” section, enter your conflict of interest statement in the “Confidential to Editor” section, and submit your "Accept" recommendation.

Reviewer #1: All comments have been addressed

2. Is the manuscript technically sound, and do the data support the conclusions?

Reviewer #1: Partly

3. Has the statistical analysis been performed appropriately and rigorously? 

Reviewer #1: Yes

4. Have the authors made all data underlying the findings in their manuscript fully available?

Reviewer #1: Yes

5. Is the manuscript presented in an intelligible fashion and written in standard English?

Reviewer #1: Yes

6. Review Comments to the Author

Reviewer #1: Dear authors

I commend you for the the study and the manuscript, and thank you for the changes made.

I just have two further comments:

1) In the first two paragraph of the introduction, the definition of ischemia and blood flow restriction is still confusing.

2) In line 239-240: I am not sure I agree with the conclusion that strength-endurance is decreased based on the shorter time under tension in ischemic intra-conditioning condition. The number of repetitions did not change. I like the discussion that ischemia may have increased eccentric velocity. Is time under tension a better marker of strength endurance compared to number of repetitions?

7. PLOS authors have the option to publish the peer review history of their article (what does this mean?). If published, this will include your full peer review and any attached files.

Reviewer #1: No

---

## [Author Response · Author response to Decision Letter 2]

5 Dec 2022

Reviewer #1: Dear authors

I commend you for the the study and the manuscript, and thank you for the changes made.

Reply -Thank you again for your valuable comments. Personally, the process of reviewing this article has taught me a lot and allowed me to look a bit differently at the colloquial or misleading terms I often use in my manuscripts.

Thank you!

I just have two further comments:

1) In the first two paragraph of the introduction, the definition of ischemia and blood flow restriction is still confusing.

Reply - in order to avoid any confusion, I have decided to completely remove the term blood flow restriction. Additionally, the first two sentences were re-written. (L55-57)

2) In line 239-240: I am not sure I agree with the conclusion that strength-endurance is decreased based on the shorter time under tension in ischemic intra-conditioning condition. The number of repetitions did not change. 

Reply - yes, I have to admit that it may be too speculative to determine the decline in strength- endurance based on TUT alone. Thank you for this comment and I made the appropriate change by introducing time of effort instead strength-endurance performance. (L239)

I like the discussion that ischemia may have increased eccentric velocity. Is time under tension a better marker of strength endurance compared to number of repetitions?

Reply - For several years, in each of my experimental protocols, I independently analyzed the number of performed repetitions and TUT. There are indications that the TUT is a more reliable indicator of volume compared to the number of repetitions (PMID: 34043184; 32735429; 29922395; 32269656; 31817252). The main aspect that may affect differences between TUT and REPS is the fact that the repetition is considered completed only when the full movement, full repetition is perform (eccentric and concentric phases), while the TUT determines the duration of the effort and even if the effort is stopped in the middle of the entire repetition, this time is counted to the value of TUT, but this effort (not full repetition) is not counted to the number of repetitions. Therefore, it is worth analyzing both parameters.

Best Regards

Michal Wilk

---

## [Editor Report · Decision Letter 3]

26 Dec 2022

The effects of ischemia during rest intervals on strength endurance performance

PONE-D-21-36446R3

Dear Dr. Wilk,

We’re pleased to inform you that your manuscript has been judged scientifically suitable for publication and will be formally accepted for publication once it meets all outstanding technical requirements.

Kind regards,

Daniel Boullosa

Academic Editor

PLOS ONE
---

## [Editor Report · Acceptance letter]

28 Mar 2023

PONE-D-21-36446R3 

The effects of ischemia during rest intervals on strength endurance performance 

Dear Dr. Wilk:

I'm pleased to inform you that your manuscript has been deemed suitable for publication in PLOS ONE. Congratulations! Your manuscript is now with our production department. 

Kind regards, 

on behalf of

Dr. Daniel Boullosa 

Academic Editor

PLOS ONE